# Change in the Mindset of a Paediatric Exercise Physiologist: A Review of Fifty Years Research

**DOI:** 10.3390/ijerph17082888

**Published:** 2020-04-22

**Authors:** Han C. G. Kemper

**Affiliations:** Emeritus Professor in Health and Physical Activity, Amsterdam UMC, Amsterdam Public Health Research Institute, 1081 HV Amsterdam, The Netherlands; hancgkemper@upcmail.nl

**Keywords:** physical activity, adolescents, aerobic power, bone density, randomized control trials, longitudinal designs

## Abstract

In this review, the career of a pediatric exercise physiologist (HCGK) is given over a period of almost 50 years. His research was concentrated on the relationship of physical activity (physical education, sport, and daily physical activity) with health and fitness in teenagers in secondary schools. (1) His first experiment was an exercise test on a bicycle ergometer to measure aerobic fitness by estimating physical work capacity at a heart rate of 170 beats/minute (PWC_170_). (2) Secondly, a randomized control trial (RCT) was performed with an intervention of more intensive physical education (PE) with circuit interval training during three lessons per week over a period of six weeks. (3) Thereafter, a second RCT was performed with an intervention of two extra PE lessons per week over a whole school year. The results of these two RCTs appeared to be small or nonsignificant, probably because the effects were confounded by differences in maturation and the habitual physical activity of these teenagers. (4) Therefore, the scope of the research was changed into the direction of a long-term longitudinal study (the Amsterdam Growth And Health Longitudinal Study). This study included male and female teenagers that were followed over many years to get insight into the individual changes in biological factors (growth, fitness, obesity, hypercholesterolemia, and hypertension) and lifestyle parameters such as nutrition, smoking, alcohol usage, and daily physical activity. With the help of new advanced statistical methods (generalized estimating equations, random coefficient analysis, and autoregression analysis) suitable for longitudinal data, research questions regarding repeated measurements, tracking, or stability were answered. New measurement techniques such as mineral bone density by means of dual-energy X-ray absorptiometry (DEXA) showed that bone can also be influenced by short bursts of mechanical load. This changed his mind: In children and adolescents, not only can daily aerobic exercise of at least 30 to 60 min duration increase the aerobic power of muscles, but very short highly intensive bursts of less than one minute per day can also increase the strength of their bones.

## 1. Introduction

Almost 50 years ago, a pediatric exercise physiologist (HCGK) started his career as a teacher of physical education (PE) in Amsterdam (Ignatius College), and, at the same time, he worked on his PhD thesis as an exercise physiologist at the University of Amsterdam (Jan Swammerdam Institute). His dream was to demonstrate the importance of daily PE lessons at school for the health of teenagers. Because he was a teacher in PE at a boys’ school in the morning and, in the afternoon, a scientist at the University of Amsterdam (UvA), this combination could realize his dream. 

In this article, the course of a career of a scientist is described over a period of 50 years in which he worked at the Amsterdam UMC Public Health Research Institute (former AMC at the UvA and VUMC of the VU) in Amsterdam, The Netherlands. 

In the 1960s, exercise physiology in children and teenagers was virtually scarce or non-existing in international literature. Only a few exercise physiologists from the Scandinavian countries (Asmussen, Astrand, Ekblom, Eriksson, and Hermansen), Germany (Ilmarinen, Mocellin, and Rutenfranz), Canada (Bailey, Shephard, and Bar-Or) and the USA (Montoye, Cumming, and Rowland) were interested in the exercise and training of adolescents. These scientists supported the development of pediatric exercise research in adolescents, as presented in this review. The majority of them were participants of the European Group of Pediatric Work Physiology who have had meetings since 1968 [1]. What is new is that the described experiments were integrated into the school environment (less self-selection), and not only in male teenagers but also female teenagers were included.

The golden thread in the four presented individual studies is the role of physical activity in school-based PE in teenagers. Starting with testing male teenagers on a specially designed racing bicycle ergometer (*n* = 80), two intervention studies were performed: one with more intensive PE lessons with interval circuit training three times per week over a six week period (*n* = 45) and one with two extra lessons per week (five instead of three) of PE over a whole school year (*n* = 70). The last study included a longitudinal study aimed at measuring individual changes in health and lifestyle in male and female teenagers from two schools (*n* = ca. 600) that were followed over many years in their growth, health, physical fitness with a special interest in the change in their physical activity pattern measured with physical activity interviews, step counters, and heart rate monitors.

### 1.1. Heart Rate during Bicycle Ergometer Exercise in Watts/Kg Bodyweight in 12- and 13-Year-Old Boys

In a group of 80 boys with the same calendar age (12–13 years), but with differences in body weight (30 to 55 kg), heart rate was measured during three discontinuous series of increasing loads (1.0, 1.5, and 2.0 watt/kg). Each period lasted 6 min, with relief intervals of 1 min in between (see Figure 1).

The results showed that during the last 2 min of each of the three workloads, no constant heart rate (steady-state) could be established (significant at *p* = 0.001). Between the six weight groups, no significant differences (*p* > 0.05) in rest, the three workloads, or the recovery periods between the workloads could be proved [2]. The physical workload capacity at a heart rate of 170 beats/minute (PWC_170_) calculated from the regression equation between heart rate and workload/kilogram body weight was 2.0 watt/kg and the lowest mean value compared to other studies with boys of the same age (see Table 1).

This study was also published in a popular Dutch journal (Ad Valvas, Amsterdam), and the heading of this article was clearly formulating the conclusion: “Each teenager has a heart that is tailor-made”.

### 1.2. Two Randomised Control Trials (RCT’s)

Two quasi-randomized control trials (RCT) in 12–13 years old schoolboys were executed with the independent variable of the intensity of the PE lessons or the frequency of the PE lessons a week. We call it semi-RCTs because the school classes were not randomized on an individual base.

The first RCT was a six-week three-times per week PE program with 15 min of an intensive interval circuit training in the experimental class (*n* = 22), compared with a control class (*n* = 23) without interval circuit training [3].

We started to measure the health and fitness effects by measuring simple anthropometric measurements such as height and weight, skinfolds, and circumferences. Physiologic measurements such as forced expiratory ventilation in 1 sec (FEV_1_) and bicycle workload per kg body weight at a heart rate of 170 beats/min (PWC_170_) were also included [4].

The published results were ambivalent: In the six-week experiment, no change in morphological characteristics could be proved, but the interval circuit training resulted in a significant improvement in the forced expiratory ventilation. Also, an increase in body height and hip-width could be established in these 12–13-year-old boys over this short period. Comparable results were found by the Belgian group from the University of Gent by Vrijens and van Uytvanck, published in 1969 [5].

The second RCT evaluated the effects of a 5- versus 3-lessons-a-week PE program during a whole school year (nine months). Two control classes were given the usual number of 3 lessons per week (*n* = 33), and they were compared with two experimental classes with 5 lessons per week (*n* = 37) [6].

In this pretest-posttest design, the dependent variables were measured at the beginning and at the end of the school year. To reduce the number of dependent variables, the grouped pre-test data were factor analyzed (varimax rotation). Out of the factors that could be discriminated, the most representative variables, with identical factor structure, were used to formulate eight hypotheses:Total body fat (%) diminishesCorrected upper arm diameter (mm) increasesHandgrip (kg) increasesBent arm hang (sec) increases50 m shuttle run (sec) diminishesPlate tapping (sec) decreasesPhysical working capacity (PWC_170_ in watt/kg body weight) increasesForced expiratory volume (FEV%) increases

The effects on these eight health-related fitness measurements were analyzed by co-variance and corrected for biological age (by X-ray of hand/wrist bones) and by daily physical activity (pedometer week scores). 

Only handgrip force appeared significantly increased (*p* < 0.05) by the two extra PE lessons after the school year compared with the control group (see Table 2). 

In this one school year experiment with two extra lessons, however, the skills learnt in the lessons (gymnologic tests), increased significantly more (*p* < 0.009) in the experimental group than in the control group (see Figure 2). 

The most recent study from Slingerland et al. in 2014 about the contribution of physical education to levels of physical activity in children and adolescents showed that intensive lessons of PE can support the activity levels of pupils as long as PE lessons are given at least five days per week [7].

### 1.3. Multiple Longitudinal Study

Thereafter, we started a multiple longitudinal study (the Amsterdam Growth and Health Longitudinal Study (AGAHLS)) to measure the changes in growth, maturation, physical fitness (neuromuscular and aerobic) in both boys and girls (age 12 to 13 years) from two secondary schools (in Amsterdam and Purmerend) over a period of 4 years during their school education [8]. The second school (Purmerend) was only measured once in the four years and used as a control for the effect of repeated measurements (see Figure 3).

To keep the dropout rate as small as possible, the measurements took place in front of the schools in a mobile laboratory (10 × 3 × 2 m). This combined an exercise unit (with a treadmill) and rooms for anthropometry, and for interviewing (see Figure 4).

Maximal aerobic fitness was measured by a direct method using a progressive protocol on a treadmill: running at a constant speed of 8 km/h with increasing slopes [9].

One of the results showed that both boys’ and girls’ maximal aerobic fitness (VO_2_max in L/min) was increasing during this adolescent period, but more in boys than in girls (upper part of Figure 5) [10]. However, VO_2_max per kg body weight over the same age period remained the same in boys and decreased in girls (lower part of Figure 5). The interrupted lines are the results of Astrand from Swedish male and female adolescents in 1952 [11]. The mean values of this 25-years older cross-sectional study are comparable with the AGAHLS data. 

Also, daily physical activity (during school hours, in leisure time, and all other in- and outdoor activities) was monitored by heart rate, pedometers, and physical activity interviews). Results showed that the more active boys and girls showed higher VO_2_max values than the less active ones. However, the aerobic power was not influenced by this negative trend in lifestyle (see Figure 6).

### 1.4. Advanced Statistical Methods

In longitudinal studies, the data collection of repeated measurements in the same participants over the years of follow-up (so-called pure longitudinal data) can be used for analyses of individual changes in the health and lifestyles.

However, these data are not independent, and, therefore, simple statistical analyses such as (M)ANOVA cannot be used. Therefore, in our AGAHLS, with eight repeated measurements over a period of 25 years of follow-up (age 12 till age 36), we had to use new and more advanced statistical methods: generalized estimating equations (GEE analysis) and random coefficient analysis have been implemented on the data [12].

To answer the question of whether there is an association between maximal aerobic fitness (peak oxygen uptake/BW) and physical activity (questionnaire/interview), we analyzed pure longitudinal data [13] and, later, also mixed longitudinal data with 1194 data points in males and 1356 in females with GEE [14]. Vigorous physical activity (VPA), with an intensity of more than 10 times the basal metabolic rate (>10 MET), revealed a positive and significant association in both males and females with the maximal aerobic fitness over the range from 12 to 36 years. In Figure 7, the mean values of peak oxygen uptake/BW, and, in Figure 8, the mean vigorous physical activity in males and females over the 25-year follow-up is shown.

The longitudinal relation between VPA (METs/week) and maximal aerobic fitness (VO_2_max/kg body weight) over the age range from 12 to 36 years revealed a positive and significant (*p* < 0.001) association. This, in contrast with studies stretching back over 35 years, which showed no meaningful relationship [15,16].

The difference with the current study is that many of these studies did not monitor VPA over a sufficiently long duration and that our data extend over a longer age range (almost 25 years), including young adulthood. The limitation of our physical activity data is that they came from a standardized interview, based on a questionnaire [17] and not on objective physical activity measurements such as pedometry and heart rate monitoring. The last two methods could only be used during the first four years during adolescence. However, during the teenage period, we checked the reliability and the validity of our physical activity questionnaire (PAQ); the test–retest reliability was high (r =0.62 in boys and r= 0.75 in girls). The validity compared with a pedometer and 48-hr heart rate registration was relatively low (r =0.16 and 0.20, respectively) [18]. 

### 1.5. Changes in Bones by Mechanical Load

This AGAHLS was continued, and there was a 15-year follow-up till the age of 42 years. In the meantime, new and non-invasive measurement methods were developed, such as bone density measurements by dual-energy X-ray absorption (DEXA).

As an exercise physiologist, HCGK was never interested in the health of bones because he assumed that the human bone is a more or less inert tissue that is not well supplied by blood. Animal experiments with swimming rats with backpacks and roosters with loaded wings showed, however, that their bones became thicker (higher density) and showed a better trabecular architecture. Moreover, the most effective physical activity was not a long duration with a relatively low load (walking and jogging) but appeared to be a high mechanical load of relatively short duration [19].

In the follow-up, measurements of bone density of the hip, lumbar spine, and wrist of our subjects were also included. The physical activity data from questionnaires (PAQ) collected over a period of 15 years were related to the development of bone density of the youngsters. It could be clearly shown that short mechanical, physical activity (such as sprinting and jumping) experienced during a 15-year period explained the amount of bone density significantly better than metabolic long-duration physical activities (such as long-distance running)) [20].

This result changed the mindset of the pediatric exercise physiologist (see Figure 9) [21]. From the longitudinal data of AGAHLS, it can be concluded that short high-intensity weight-bearing physical activities are very important for normal growth and development of the skeleton in youth. While for an increase in aerobic fitness at least one hour of daily metabolic physical activity (jogging, running, or walking) is needed, for increasing bone health only one minute (six times of 10 seconds spread over the day) of a high mechanical load seems to be enough (for example, rope skipping and climbing stairs up and down). 

### 1.6. Other Recent Longitudinal Studies

As a result of the Amsterdam Growth And Health Studies, young investigators such as Kotsedi Monyeki and Andries Monyeki from South Africa and Romulo Fernandes from Brazil have taken up new research projects in their relatively poor countries, with adolescents from other races and sometimes in a poor environments, to investigate growth and lifestyle on health.

The Physical Activity and Health Longitudinal (PAHL) study from A Monyeki et al. [22], the Ellisras Longitudinal Study (ELS) from K Monyeki et al. [23], and the Analysis of Behaviours of Children During Growth (ABCD-Growth study) [24] from Fernandes are three examples of recent longitudinal studies set up in other countries with new and modern techniques. They show that pediatric exercise science will bloom in the near future.

## 2. Conclusions

Our results show that the effects of PE are dependent on the individual lifestyle of the teenagers: In highly active boys and girls, more intensive or extra lessons PE add only 5%–10% of their total daily activity per week (bicycle to and from school, sporting in clubs) compared to their peers who are physically inactive (no sports, sitting at home, and using passive transport).

The Amsterdam Growth And Health Longitudinal study reveals the effects of growth and lifestyle in both male and female adolescents over a period of almost 25 years. Advanced statistical analyses showed the limited effects of aerobic physical fitness in this young adult population.

In a recent publication of the Health Council of the Netherlands (Gezondheidsraad), with respect to guidelines for the lifestyle daily physical activity, the following statements were made [25]:Physical activity is good; more movement is better;At least 150 min per week of moderate physical activity (walking, bicycling);At least twice a week muscle- and bone-strengthening exercises;Prevent long-time sitting.

These movement guidelines from 2017 are partly based on our research.

## Figures and Tables

**Figure 1 ijerph-17-02888-f001:**
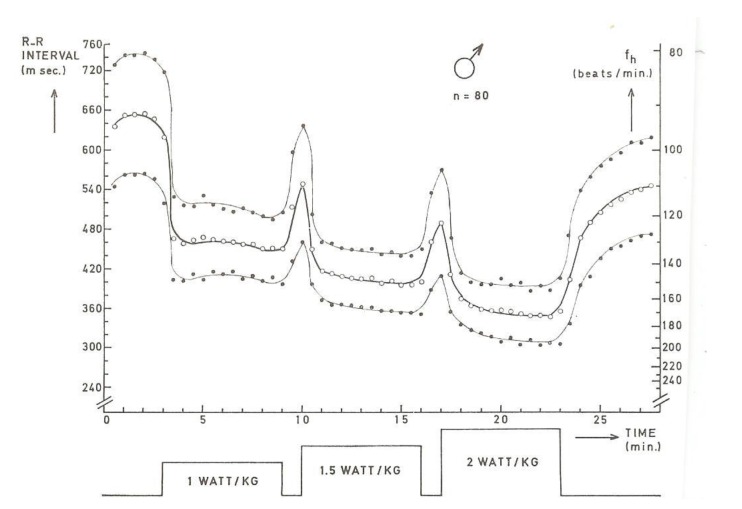
Mean (open symbols) and standard deviation (black symbols) of the heart rate (beats /min) of all subjects for every 30-s period [2].

**Figure 2 ijerph-17-02888-f002:**
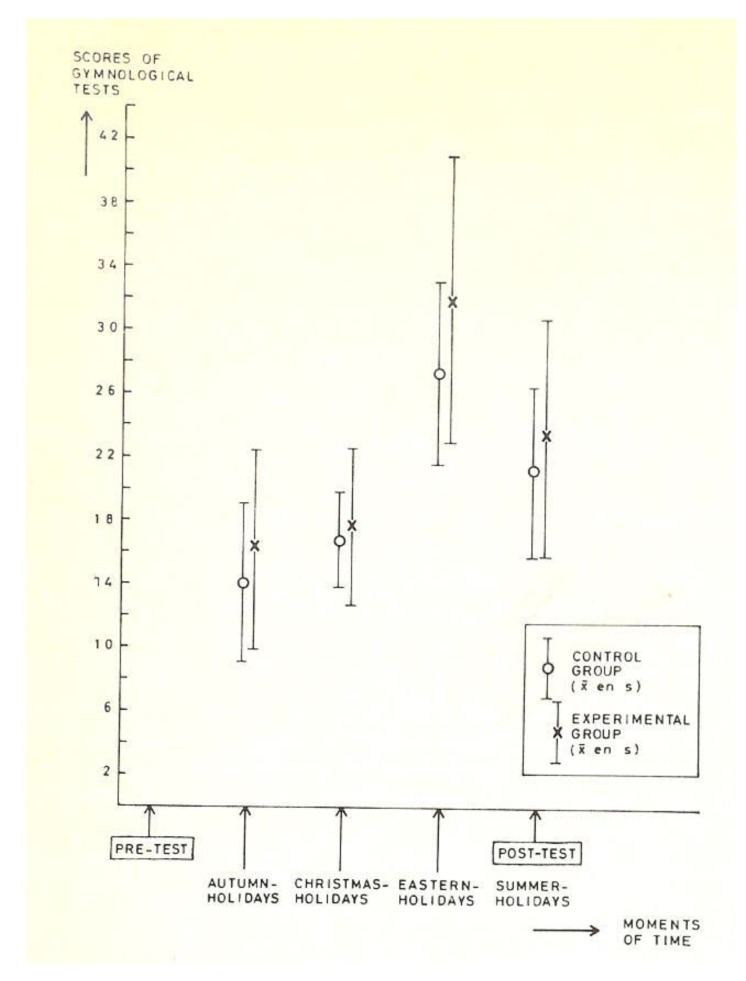
Mean and standard deviation of gymnologic scores of experimental groups (black crosses) and control group (open symbols) during the four periods of measurement [6].

**Figure 3 ijerph-17-02888-f003:**
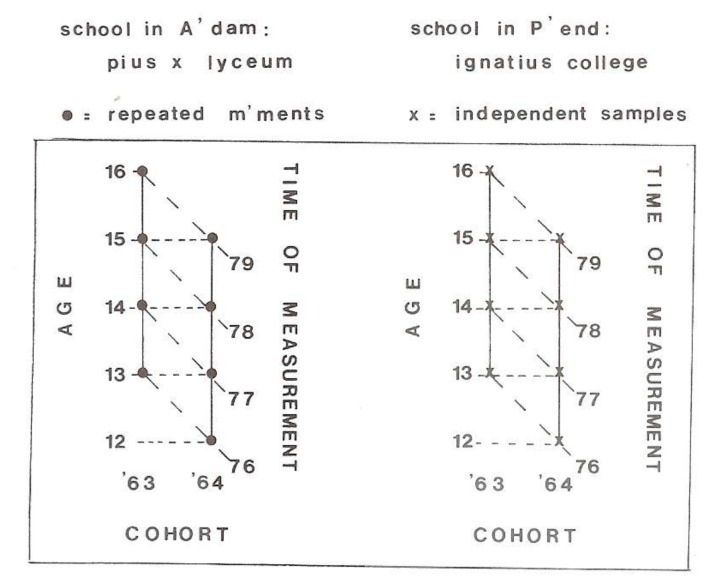
Multiple longitudinal design of Amsterdam Growth And Health Longitudinal Study (AGAHLS) in the two schools: Pius X Lyceum in Amsterdam (two birth cohorts (1963 and 1964) with repeated measurements) and Ignatius College in Purmerend (two birth cohorts (1963 and 1964) with independent samples) [8].

**Figure 4 ijerph-17-02888-f004:**
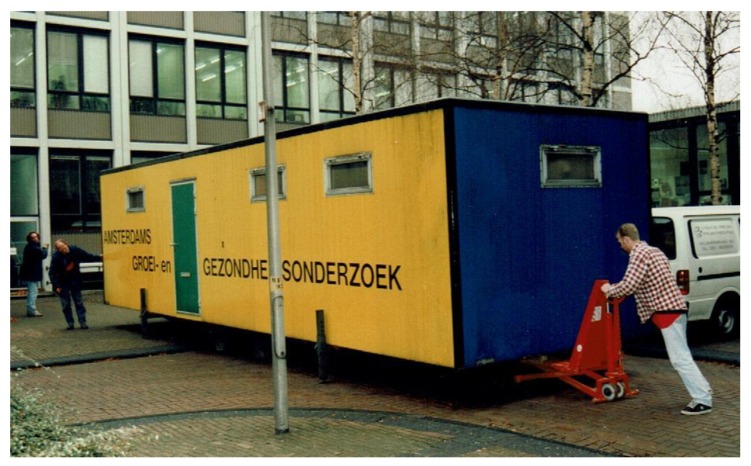
The mobile laboratory ready for transport to one of the two secondary schools.

**Figure 5 ijerph-17-02888-f005:**
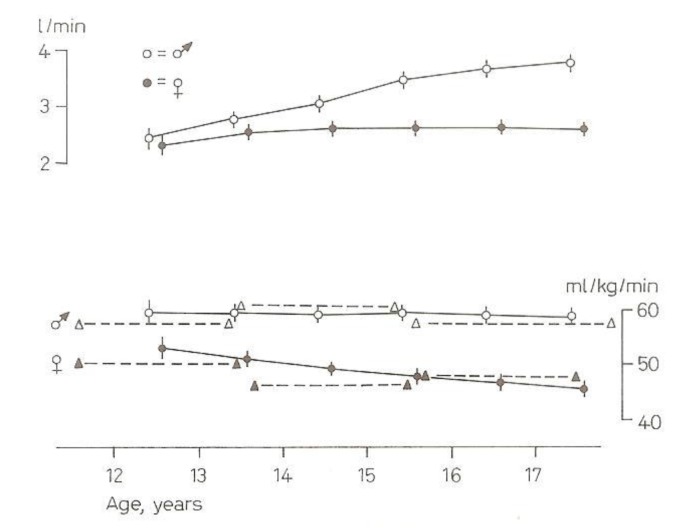
Mean and standard error of absolute maximal aerobic fitness (VO_2_max in L/min) (upper part) and relative maximal aerobic fitness (VO_2_max/BW (mL/kg/min) (lower part) in boys and girls versus calendar age [10]. Interrupted lines indicate the results of Astrand [11].

**Figure 6 ijerph-17-02888-f006:**
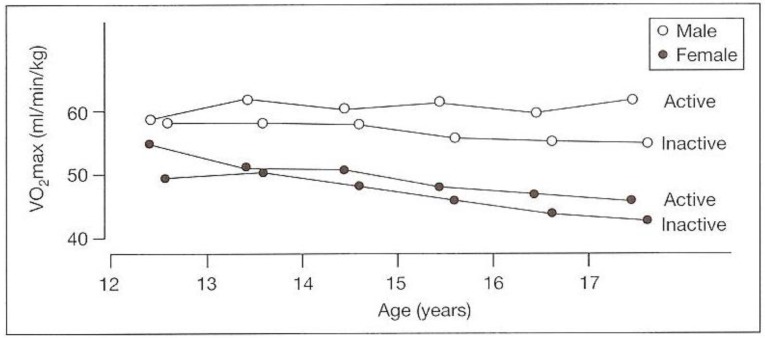
Development of VO_2_max per kg body weight in boys and girls with relatively high (active) and low (inactive patterns during the adolescent period of 12–17 years of age) [10].

**Figure 7 ijerph-17-02888-f007:**
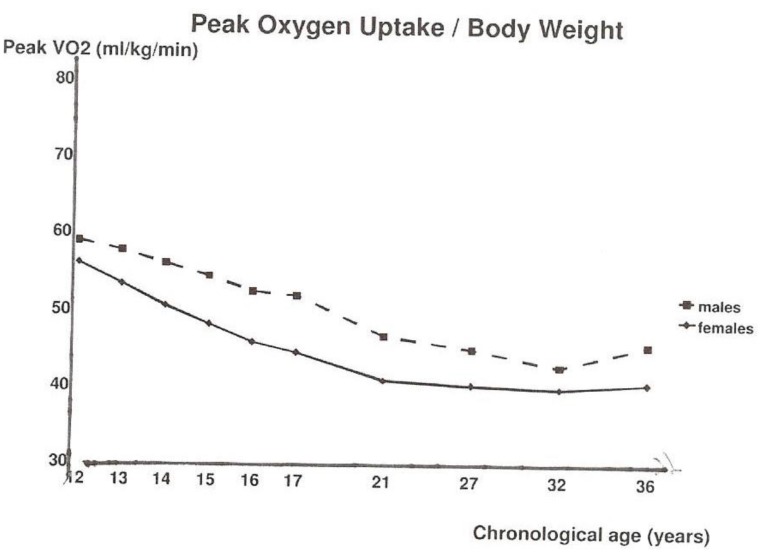
Mean values of peak oxygen uptake relative to bodyweight separately for males and females [14].

**Figure 8 ijerph-17-02888-f008:**
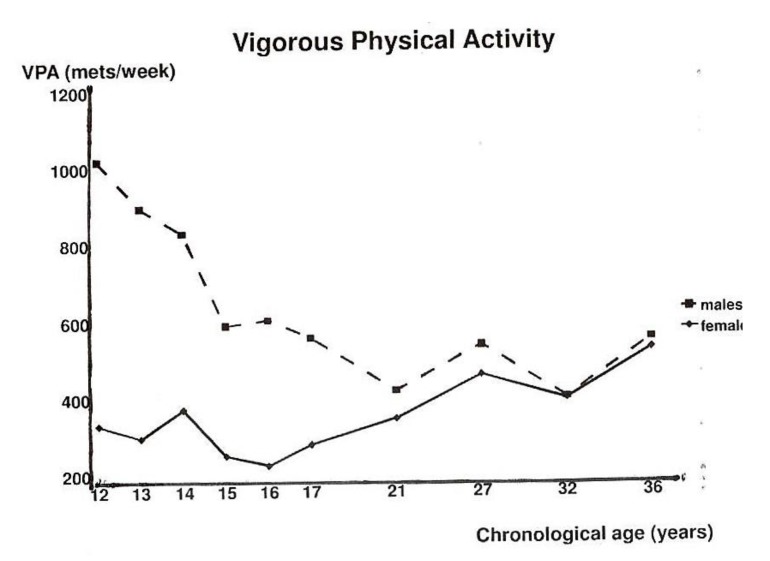
Mean values of vigorous physical activity levels (VPA) in METs/week, separately for males and females [14].

**Figure 9 ijerph-17-02888-f009:**
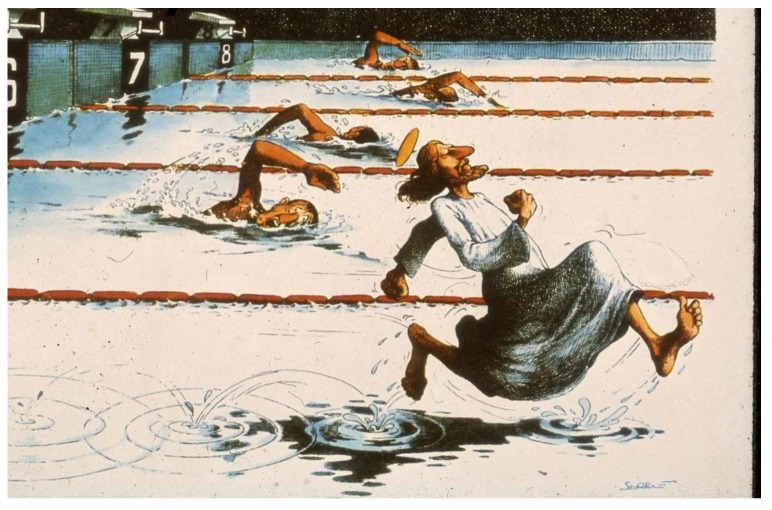
Change in mindset according to Serré [21].

**Table 1 ijerph-17-02888-t001:** Comparison of PWC_170_ values (=W_170_ (W/kg)) of the 13-year-old boys with boys of the same age in seven other countries [2].

		*n*	Weight (Kg)	W_170_ (W/kg)
**California**	(Adams, 1961)	60	48	2.4
**Stockholm**	(Adams, 1961)	36	42	2.4
**Turku, Finl.**	(Olavi, 1965)	27	36	2.7
**Canada**	(C.A.H.P.E.R., 1968)	410	42	2.2
**Praag**	(Mᾴcek, 1970)	90	46	2.6
**Japan**	(Ishiko, 1971)	22	40	2.3
**Giessen, W. Germ**	(Mocellin, 1971)	21	42	2.3
**Amsterdam**	(Kemper, 1971)	80	42	2.0

**Table 2 ijerph-17-02888-t002:** Analysis of covariance of the eight hypotheses [6].

	Test 1	Test 2	Test 3
FEV%	n.s.		
Corr. upp. arm diam.	*	n.s.	
FAT%	n.s.		
Plate tapping	n.s.		
50 m shuttle run	n.s.		
Bent arm hang	n.s.		
Handgrip	**	*	n.s.
W_170_	n.s.		

* *p* ≤ 0.05; ** *p* ≤ 0.01; n.s. = not significant *p* > 0.05.

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
