# Peer review of "Change in the Mindset of a Paediatric Exercise Physiologist: A Review of Fifty Years Research"

_ijerph, 2020, doi:10.3390/ijerph17082888_

Round 1
Reviewer 1 Report
Formal:
-there should be no personalized words (f.i. he, she etc.)
-no points after 1, 1.1 etc.
-no (...) within (…)
Literature:
-more comparative literature
Author Response
I prefer not to repeat HCGK every time, and also "his" is difficult to change.
I leave the points after 1 , 2.2 and no ... within..
Comparative research is added.
Reviewer 2 Report
The article intends to present the change in the mindset of a physiologist working in the field of physical education. The article is included in the review section. However, the data are related to a single person experience. The scientific reseach is very poorly presented.
Introduction. The introduction is described as a personal experience. The rigorous scientific data are missing.
Material and methods. The section head is missing. The methodology of the review is not included. There are presented only two randomised controls trials whose author is the author of the article himself.
The discussion section is missing.
The conclusions are briefly presented. They include recommendations for the physical activity in general, without being related to the current work.
The reference section must be improved. 7 of the 12 references are belonging to the author of the present article.
Author Response
I call it a review about my own scientific work in the field of physical education of teenagers.
The formel section heads as methodology and discussion are not used because they are not applicable in this paper.
In the introduction there is no need of scientific data.
I shall include more references from others.
Reviewer 3 Report
The article is a mosaic of various scientific research conducted by the Author. I miss connecting these parts with each other. Research is conducted based on various methodologies, which are very briefly described. The analyzed research are valuable, so it is worth describing them in more detail.
A more detailed description of the experimental factor is needed - how did the tested classes differ. Or just the number of physical education lessons per week? How many people took part in the experiment described?
Lack of research methodology carried out among 1194 men and 1356 women - what questionnaire was used, was it verified, was it reliable?
Table 3 presents analysis of covariance of eight hypotheses. Nowhere in the article, however, no hypotheses appear. This needs supplementing.
The conclusions at the end of the thesis are not correlated with what the Author writes about in the presented work. The recommendations are generally known and are not related to the results of the described research.
In this form I do not recommend the article for publication. I am asking for rewording and completing. It is also worth adding a little more contemporary references.
Author Response
I appreciate the comments on my article and will connect the different research papers with a golden thread as suggested by another reviewer.
Methodology will be more explained as well as the differences in the experimental factor in the RCT's.
Also the hypotheses will be made more explicitely.
More references will be added of comparable research.
Conclusions from the presented results are extended.
Reviewer 4 Report
General comments:
I think this could be a very interesting summary of some excellent research that has been conducted over the past 40+ years. The author has clearly made a real impact to the paediatric exercise physiology field as demonstrated by the fact that current PA guidelines in the Netherlands are largely based on their research findings. In the current review, the individual studies have been described quite well, but the link between them has not been explained in full. A longer introduction may help to place this review and its chronological studies into a broader context. Additionally, the link between individual studies and the ‘golden thread’ connecting them together could be provided and explained throughout. For example, how did the first study inform (if at all) the design of two RCTs? Similarly, in the second RCT significant difference was only found in handgrip strength, so it would be interesting to read the author’s explanation as to why other seven measurements were not significantly affected with two extra PE classes per week. Again, did those findings inform their longitudinal studies designs and health/fitness measurements taken? Sharing reflections on the journey with the reader would be very helpful. Therefore, I would find it much more interesting if it was written as a reflective account detailing the challenges the author had experienced as a researcher and PE teacher over time. In other words, the current review should be greater than the sum of its parts. Whilst there was an element of this later in the review where the author outlined the decision to start using DEXA scanner to test the effects exercise/PA on bone density, the whole narrative should be written in that manner. This would also nicely support the chosen title.
Specific comments:
Line 34: Please change ‘boy’s’ to ‘boys’’
Line 59: Consider moving this sentence after the next paragraph (lines 60-62) because it only states that the published results were ambivalent, but they are not actually presented until further down.
Line 67: Please change ‘significant’ to ‘significantly’.
Line 90 (and thereafter): VO2max should be ‘V̇O2max’
Hope you find my comments helpful.
Author Response
I am thankfull for the positive comments on my paper.
I will write a longer introduction to place the individual studies in a broader context and connecting them with each other.
I try to write it more as a reflected account with the challenges experience by me during the 40+ years of research.
Round 2
Reviewer 3 Report
There is still no information on the methodology of the research carried out among 1194 men and 1356 women. The author did not answer my questions: what questionnaire was used, was it verified, was it reliable?
In the corrected passage regarding Table 3, it is still not known what hypotheses the author of the work is about.
The references has been supplemented with 4 items from 1948, 1969, 1975 and 2000, so they are still not the latest publications on this topic - this attention has not been corrected.
The corrected conclusions are only partly related to the content of the article and the discussed research.
In this form I do not recommend the article for publication. In my opinion, this will be possible after correcting the previous comments, which I wrote in the first review.
Author Response
Thanks for your second review.
The numbers of men and women are not numbers of participants, but numbers of measurements (over the ten measurement years) during the whole longitudinal study. The methodology is called mixed longitudinal: some participants has 10 and others one, two or more points of measurements; all are taken in to account.
The physical activity questionnaire is validated against pedometry and heart rate monitoring, this is included in the text.
I have formulated the 8 hypotheses about the effect of two extra lessons PE experiment in text before the table 3.
I have created a new paragraph 2.6 with latest publications from recent ongoing longitudinal studies.
Conclusions related to the content of the article are extended.
Reviewer 4 Report
General comments:
Firstly, thank you for taking my comments on board. I do think that the additional information presented has made the manuscript easier to follow and more interesting to read. However, I still feel that some important details are missing. For example, the 3rd paragraph (Lines 47-52) in Introduction could be expanded by stating what research was being performed by those few scientists around the world. In other words, what did the author do new/differently to the rest of them and why? From the details presented in Lines 54-55 it is still not very clearly whether the same sample of teenagers was used in the initial HR testing study and the two RCTs. Judging from the participants’ age I assume it was the same group of participants, but this could be explained better. Again, this could help you link the papers more coherently. Finally, whilst some other studies have been added for comparison purposes, results could be discussed in greater detail. E.g. why frequency of PE appears to be more important than intensity of PE classes (difference between the two RCTs)?
Specific comments:
If raw data are available, could all graphs be reproduced so they look the same? Currently, it doesn’t look very professional.
Line 12: Change ‘on’ to ‘in’.
Line 15: Please check grammar.
Lines 16-18: This sentence is incomplete.
Line 62: No space after 12.
Line 139: Check grammar.
Line 196: What is meant by ‘pure longitudinal’?
Author Response
Thanks for the reply.
In the introduction I added what is new and different compared to the few experienced pediatric exercise scientists I learned from.
The samples in the studies have the same calendar age and are from the same school, but are not the same because they are cohorts from a different birth year.
The two RCT's are not only different in the Intensity (circuit interval training in PE lessons and frequency (3 versus 5 PE lessons per week) but also in the length of the experiments (6 weeks or 9 months). I will make this more clear in the manuscript.
The editor advised me to use the old figures and tables.
The specific comments (english, spaces and grammar)are taken in consideration and changed in the text.
"Pure longitudinal" means only data from participants that completed all the repeated measurements over the years